**∂ | Open Peer Review** | Bacteriology | Research Article

# Uncovering the boundaries of *Campylobacter* species through large-scale phylogenetic and nucleotide identity analyses

Ruochen Wu,[1] Michael Payne,[1] Li Zhang,[1] Ruiting Lan[1]

**ABSTRACT** *Campylobacter* species are typically helical shaped, Gram-negative, and non-spore-forming bacteria. Species in this genus include established foodborne and animal pathogens as well as emerging pathogens. The accumulation of genomic data from the *Campylobacter* genus has increased exponentially in recent years, accompanied by the discovery of putative new species. At present, the lack of a standardized species boundary complicates distinguishing established and novel species. We defined the *Campylobacter* genus core genome (500 loci) using publicly available *Campylobacter* complete genomes (*n* = 498) and constructed a core genome phylogeny using 2,193 publicly available *Campylobacter* genomes to examine inter-species diversity and species boundaries. Utilizing 8,440 *Campylobacter* genomes representing 33 species and 8 subspecies, we found species delineation based on an average nucleotide identity (ANI) cutoff of 94.2% is consistent with the core genome phylogeny. We identified 60 ANI genomic species that delineated *Campylobacter* species in concordance with previous comparative genetic studies. All pairwise ANI genomic species pairs had *in silico* DNA-DNA hybridization scores of less than 70%, supporting their delineation as separate species. We provide the tool *Campylobacter* Genomic Species typer (CampyGStyper) that assigns ANI genomic species to query genomes based on ANI similarities to medoid genomes from each ANI genomic species with an accuracy of 99.96%. The ANI genomic species definitions proposed here allow consistent species definition in the *Campylobacter* genus and will facilitate the detection of novel species in the future.

**IMPORTANCE** In recent years, *Campylobacter* has gained recognition as the leading cause of bacterial gastroenteritis worldwide, leading to a substantial rise in the collection of genomic data of the *Campylobacter* genus in public databases. Currently, a standardized *Campylobacter* species boundary at the genomic level is absent, leading to challenges in detecting emerging pathogens and defining putative novel species within this genus. We used a comprehensive representation of genomes of the *Campylobacter* genus to construct a core genome phylogenetic tree. Furthermore, we found an average nucleotide identity (ANI) of 94.2% as the optimal cutoff to define the *Campylobacter* species. Using this cutoff, we identified 60 ANI genomic species which provided a standardized species definition and nomenclature. Importantly, we have developed *Campylobacter* Genomic Species typer (CampyGStyper), which can robustly and accurately assign these ANI genomic species to *Campylobacter* genomes, thereby aiding pathogen surveillance and facilitating evolutionary and epidemiological studies of existing and emerging pathogens in the genus *Campylobacter*.

**KEYWORDS** *Campylobacter*, average nucleotide identity, genomic species, core genome

C*ampylobacter* is one of the most common human enteric pathogens in the world (1, 2). This genus encompasses established pathogens in clinical and agricultural

Address correspondence to Li Zhang, l.zhang@unsw.edu.au, or Ruiting Lan, r.lan@unsw.edu.au.

The authors declare no conflict of interest.

settings, as well as emerging pathogens. Currently, there are 33 recognized *Campylobacter* species and 9 subspecies (3–5) with various numbers of collected genomes in the public database. *Campylobacter* species are diverse and naturally colonize humans, domesticated animals (such as dogs, cats, chickens, sheep, and cattle) (1, 6), some wild animals (birds, reptiles, and shellfish), and the ecological niches they inhabit (7, 8). Genomic analyses of *Campylobacter* from different animal reservoirs have demonstrated that *Campylobacter* species achieve host-specific adaptations through horizontal gene transfer (HGT) (1, 9, 10). *Campylobacter jejuni* and *Campylobacter coli* are the main causes of campylobacteriosis, leading to around 500 million cases of acute diarrhea across the globe annually (1, 2, 9). Other human-hosted *Campylobacter* species include *Campylobacter concisus*, *Campylobacter curvus*, *Campylobacter gracilis*, *Campylobacter hominis*, *Campylobacter rectus*, *Campylobacter lari*, *Campylobacter upsaliensis*, and *Campylobacter showae*. These emerging pathogenic species are associated with gastroenteritis infections and chronic inflammatory diseases (10, 11). *Campylobacter* species are also responsible for causing enteritis, spontaneous abortion, infertility, and other illnesses in various animals. *Campylobacter fetus* is the most important cause of infertility in farm animals. The two subspecies of *C. fetus*, *venerealis* and *fetus*, can both cause bovine, ovine, and caprine abortion and bovine venereal campylobacteriosis, posing a significant economic burden on the agricultural industry (8).

Although *Campylobacter* has become increasingly important in public health since its initial description in 1963, its taxonomy has undergone substantial changes as new classification methodologies emerge (1). Currently, species determination can be performed by comparing the unspecified *Campylobacter* genomes to exact k-mer matches from the NCBI taxonomy database (https://www.ncbi.nlm.nih.gov/taxonomy) using tools such as Kraken2 (12). However, the *Campylobacter* nomenclatures in the NCBI taxonomy database are not based on a comprehensive genetic or phylogenetic analysis of the entire genus. HGT between *Campylobacter* species also poses challenges to defining the species boundaries within this genus (10, 13, 14). Techniques such as multilocus sequence typing (MLST) are compromised because the extent of recombination between species is so broad. For example, *C. jejuni* sequences can be found in *C. coli* housekeeping genes, making it difficult to differentiate the two species using MLST (14–16). Although *C. jejuni* and *C. coli* have a well-established combined cgMLST scheme available in the pubMLST database (13), other species within the genus lack a standardized definition and have primarily been defined through sequence comparison of the 16S rRNA gene (17, 18). While 16S rRNA sequencing is a well-established tool for differentiating bacterial genomes at genus and species levels, sequence variation on the 16S rRNA gene can be influenced by substitution saturation and recombination (19, 20) and is unable to differentiate between some genetically distinct species (including distinguishing *C. jejuni* from *C. coli* and *Campylobacter helveticus* from *C. upsaliensis*) (3, 19). With the large number of whole-genome sequences now available, accurate species delineation based on genome sequences should overcome these deficiencies.

Average nucleotide identity (ANI) is a measure of the genomic similarity between genomes and has been used to delineate species boundaries in bacteria and archaea (21, 22). Furthermore, population structure and species delineation based on ANI-derived distance measures correlate with the traditional species definition based on DNA-DNA hybridization (DDH) (21, 23), with a 95% ANI corresponding to 70% DNA-DNA hybridization cutoff which is the gold standard of species delineation in bacteria (23). Having a standardized species definition is crucial for understanding the population structure and accurate classification of the species in *Campylobacter* genus. In this study, we used core genome phylogenetic analysis and ANI clustering of 8,840 genomes to derive an optimal cutoff to delineate *Campylobacter* species, which we refer to as ANI genomic species, and developed an assignment tool, *Campylobacter* Genomic Species typer (CampyGStyper), for the genomic species assignment based on genome sequence.

## MATERIALS AND METHODS

### *Campylobacter* genome retrieval and processing

A total of 498 complete genome sequences of the *Campylobacter* genus were obtained from the NCBI database (https://www.ncbi.nlm.nih.gov/) on 24 March 2022. We downloaded 88,715 publicly available *Campylobacter* raw Illumina paired-end reads and species metadata from the European Nucleotide Archive (March 2022). The shovill v1.1.0 pipeline (http://github.com/tseemann/shovill) was employed to process reads using Trimmomatic, Lighter, and Fast Length Adjustment of SHort reads (Flash) and assemble them using strategic k-mer extension for scrupulous assemblies (SKESA) (24–27). Finally, reads were remapped for error correction using Burrows–Wheeler Aligner (BWA) and Pilon (28, 29). Assembled genomes within each species with outlying assembly metrics in total assembly length, contig number, N50, or the number of sequences that make up the N50 (N50n) were removed.

### *Campylobacter* genus core gene identification

Two criteria were defined for a gene to be included in the core genome. First, the gene must be present in all but one genome in the set of non-*coli/jejuni* species complete genomes. Second, the gene must not be missing in more than 5% of complete genomes of *C. jejuni* or *C. coli*. From the 498 complete genomes, 5 genomes with insufficient annotation data and 8 unspecified *Campylobacter* genomes were excluded from the core genome selection to avoid misrepresentation of species as their species status is unknown. A singleton *Campylobacter avium* genome was also excluded based on the first selection criteria. The resulting complete genome data set consists of 485 genomes representing 31 *Campylobacter* species in the NCBI taxonomy database.

We used PROKKA to locate open-reading frames from the complete genome data set. The output GFF files from PROKKA were used as input files for Roary v3.5 to define the *Campylobacter* genus core loci. Considering the diversity of the genus, Roary v3.5 was run at multiple BLASTp percentage identities to identify potential core genes. The presence of core loci in each complete genome was determined by identifying genes with ≥40% minimum BLASTp percentage identity between genomes. Core genes that are paralogous and orthologous split into multiple fragments by erroneous assemble and/or annotation were identified and subsequently removed using a previously described Python script (30).

### Phylogenetic analyses based on core genome alignments

The allele sequences of core genes were called from a total of 2,193 genomes, including publicly available complete genomes (485) and assembled genomes from non-*coli/jejuni* species and unspecified *Campylobacter* (1,708). Using the allele sequences, 2,193 *Campylobacter* core genome sequences were constructed by concatenating alignments of individual core loci. A phylogenetic tree was constructed using IQ-TREE v2.0.662 based on the core genome alignments. Branch supports were assessed with 10,000 ultrafast bootstrap replicates and visualized using GrapeTree v1.5.0 and Interactive Tree Of Life (iTOL) v6.8.1 (31, 32).

### ANI analysis and clustering

The pubMLST *Campylobacter* seven gene MLST typing scheme was used to type 56,528 *C. jejuni* and 20,500 *C. coli* genomes (https://github.com/tseemann/mlst) (33); 2,382 and 1,225 MLST STs were identified, respectively. Furthermore, 1,464 *C. jejuni* genomes and 1,165 *C. coli* genomes belonged to novel STs. For ANI comparison, a representative genome of each MLST sequence type was selected at random; all genomes with novel MLST STs were also included. The representatives of *C. jejuni* and *C. coli* were combined with 498 complete genomes and 1,708 non-*coli/jejuni* species assembled genomes; the resulting *Campylobacter* ANI data set consisted of 8,440 genomes. The

pairwise ANI comparison was generated using fastANI v1.33 and subsequently compiled into a pairwise dissimilarity matrix with an in-house python script (21). Clustering was performed using the average linkage clustering algorithm from scikit-learn package AgglomerativeClustering (34). We examined clusters formed at every cutoff ranging between 90% and 97% with increments of 0.1% to find the ANI similarity cutoff that complements the core genome phylogenetic tree. DDH distances of the complete genomes representing the ANI genomic species were compared using the GGDC web service tool (35, 36).

## Medoid genome identification

The 8,440 genomes were grouped based on their average linkage clustering cluster at 94.2% ANI. A pairwise dissimilarity matrix was generated for each cluster and used as the input for an in-house python script utilizing the Partition Around Medoids (PAM) algorithm; the medoid genome is identified when its dissimilarity to all other genomes in the cluster is minimum. The medoid genome of each cluster was identified using FasterPAM algorithm in the package Fast k-medoids clustering in Python with default setting (37, 38).

*Campylobacter* Genomic Species typer (CampyGStyper), a tool utilizing fastANI v1.33 (21), was developed for assigning *Campylobacter* genomic species to query genomes based on their ANI similarity to the medoid genomes (https://github.com/LanLab/CampyGStyper).

## RESULTS

### Genomic data sets and *Campylobacter* genus core genome

A total of 498 complete genomes and 88,715 draft *Campylobacter* genomes were downloaded from the ENA database, among which 9,993 outlier genomes were removed based on the assembly statistics within each NCBI user submitted species (Table 1).

Using 485 complete genomes and the *C. jejuni* strain NCTC 11168 as the reference (containing 1,658 genes), 602 genes initially passed the core genome selection criteria, as described in Materials and methods. Subsequently, 60 paralogous genes and a further 42 genes with overlapping regions were identified and removed. The *Campylobacter* genus core genome thus consists of 500 genes.

Two genome data sets were constructed for phylogenetic and ANI analyses (Table S1). A core genome phylogeny data set consisted of 2,193 genomes, each of which has a core genome size of 400 kbp or larger, including 485 complete genomes and 1,708 draft genomes. The core genome phylogeny data set represented 32 *Campylobacter* species, 8 subspecies, and 141 genomes from unspecified *Campylobacter* species recognized in the NCBI taxonomy database. A larger data set (8,440 genomes), referred to as ANI analysis data set, was constructed. It included all publicly available complete genomes (498), 1,708 draft genomes from non-*coli/jejuni* species, 3,846 *C. jejuni*, and 2,390 *C. coli* draft genomes that represented unique STs in the *C. jejuni* or *C. coli* MLST scheme.

TABLE 1   Quality check criteria for assembled genomes of each NCBI user submitted species[a]

| NCBI user submitted species | Genome length upper limit(bp) | Genome length lower limit (bp) | Number of contigs | N50 length (bp) | Number of contigs at 50% genome coverage |
|---|---|---|---|---|---|
| *C. jejuni* | 1,861,136.5 | 1,504,364.5 | <145.5.0 | <301,885.5 | <20.0 |
| *C.coli* | 1,933,285.0 | 1,522,269.0 | <213.0 | <276,835.5 | <25.0 |
| *C.concisus* | 2,194,114.0 | 1,655,706.0 | <65.0 | <545,572.5 | <7.0 |
| *C. fetus* | 1,995,647.5 | 1,579,587.5 | <86.0 | <475,099.5 | <8.0 |
| *C. upsaliensis* | 1,821,661.5 | 1,488,857.5 | <157.5 | <81,786.0 | <20.0 |
| *C. hyointestinalis* | 1,961,122.5 | 1,600,718.5 | <59.5 | <511,467.5 | <8.0 |
| *C. lari* | 1,673,469.5 | 1,314,705.5 | <82.0 | <259,746.5 | <11.5 |

[a]Assembly metrics of species with less than 50 publicly available genomes were plotted on histograms, and outliers were removed manually.

## Phylogenetic analysis based on core-genome alignments from 2,193 genomes

A phylogenetic tree using the phylogeny data set of 2,193 genomes was generated to examine the concordance of phylogenetic relationships with user submitted NCBI species (Fig. 1). *C. coli* and *C. jejuni* were grouped in a common clade, consistent with previous findings (13–16, 39). *Campylobacter hepaticus* has the closest phylogenetic relationship to *C. coli* and *C. jejuni*. *C. lari*, *Campylobacter insulaenigrae*, *Campylobacter volucris*, *Campylobacter subantarcticus*, and *Campylobacter peloridis* were grouped together into one clade, which is consistent with the previously established *C. lari* group (40). *Campylobacter armoricus* was close to *C. lari*, as indicated by Boukerb et al. (4). *C. fetus*, *Campylobacter hyointestinalis*, *Campylobacter iguaniorum*, and *Campylobacter lanienae*, which are non-thermotolerant *Campylobacter* species, formed a separate clade on the phylogenetic tree. *C. upsaliensis* and *C. helveticus* grouped together which is consistent with Stanley et al. (41). *C. showae* and *C. rectus* grouped together which is consistent with Thomas et al.(42). Several species that occupy unique ecological niches were distantly related to other members of the genus on the phylogenetic tree. These include *Campylobacter pinnipediorum* and *Campylobacter cuniculorum*. Emerging pathogenic species, such as *Campylobacter sputorum*, *Campylobacter ureolyticus*, *C. hominis*, and *C. gracilis*, are phylogenetically distant. The 141 unspecified *Campylobacter* genomes were sporadically distributed on the core genome phylogenetic tree.

Most of these genomes can be found in clades of established species, but there were also clades consisting of only unspecified *Campylobacter* genomes indicating novel

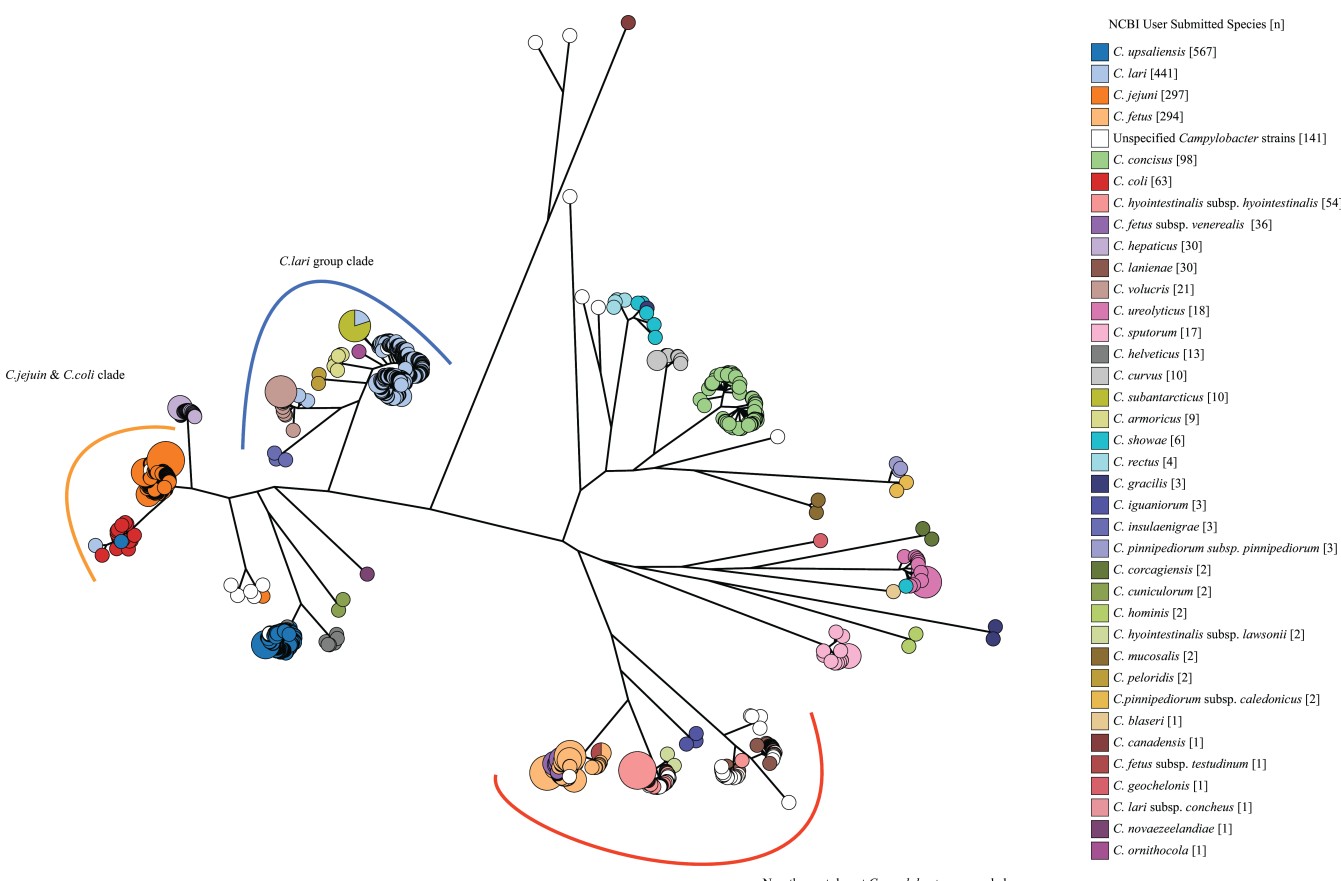

**FIG 1** Core genome phylogenetic tree of 2,193 genomes. The maximum likelihood phylogenetic tree was constructed using alignments of 500 *Campylobacter* core genes using IQ-TREE v2.0.662 and visualized using GrapeTree v1.5.0. The bootstrap value at each node was obtained from 10,000 bootstrap replicates and was all higher than 95%. Each of the genome has a core genome alignment length of ≥400 kb. The phylogenetic tree was overlaid with species metadata from NCBI SRA database.

genomic species. Genomes with a species name that does not align with their position on the phylogenetic tree indicate incorrect NCBI species assignment. Unsurprisingly, the subspecies of *C. fetus*, *C. hyointestinalis*, *C. lari*, and *C. pinnipediorum* can be found on the same phylogenetic branch of their respective species. The two genomospecies of *C. concisus* are separated into two distinct branches in the *C. concisus* clade. The branch length between the two genomospecies is longer than the branch length between established subspecies.

## Determining optimal ANI cutoff for *Campylobacter* species delineation

To determine the optimal ANI cutoff for species delineation that is consistent with the core genome phylogeny, we used the ANI analysis data set (8,440 genomes) to examine the number of clusters formed at different ANI cutoffs from 97% to 90% with every 0.1% decrement. An ANI of ≥95% is generally used as cutoff for species delineation in prokaryotes (21). For most of the *Campylobacter* species, the 95% ANI cutoff created ANI clusters that are aligned with the established species boundary. However, *C. concisus*, *C. lari*, and *C. volucris* were divided into numerous small clusters, which did not correlate with the core genome phylogeny nor with established species assignment (Fig. S1). For example, *C. concisus* was divided into 22 different clusters at ANI 95%. The subspecies of *C. hyointestinalis*, *C. pinnipediorum*, *C. fetus*, and *C. lari* would be considered as separate species at 95% ANI cutoff. The discrepancy between the ANI clustering and the core genome phylogeny indicates that the 95% ANI is too stringent to delineate *Campylobacter* genomes into the established species.

The ANI analysis data set reached a plateau of 60 clusters at ANI 94.2% and remained the same at ANI 93.8% (Fig. 2). When overlaid onto the core genome phylogenetic tree (Fig. 3), the ANI clusters were consistent with the species delineation on the core genome phylogenetic tree. The 33 established *Campylobacter* species can be identified by distinctive ANI clusters, correlating with their unique species clades on the core genome phylogenetic tree. Subspecies of *C. fetus* (*fetus* and *venerealis*), *C. pinnipediorum* (*pinnipediorum* and *caledonicus*), and *C. hyointestinalis* (*hyointestinalis* and *lawsonii*) were each grouped into distinctive, species-level ANI clusters: *C. fetus* (ANI genomic species 16), *C. hyointestinalis* (ANI genomic species 8), and *C. pinnipediorum* (ANI genomic species 9). The biovars of *C. sputorum* were grouped into a distinctive, species-level ANI clusters (ANI genomic species 5). The two genomospecies of *C. concisus* (43–45) corresponded to ANI clusters 6 and 2. The reptile associated *C. fetus* subsp. *testudinum* was grouped in ANI cluster 55 distinct from the other *C. fetus* subspecies. Respectively, ANI clusters 1 and 14

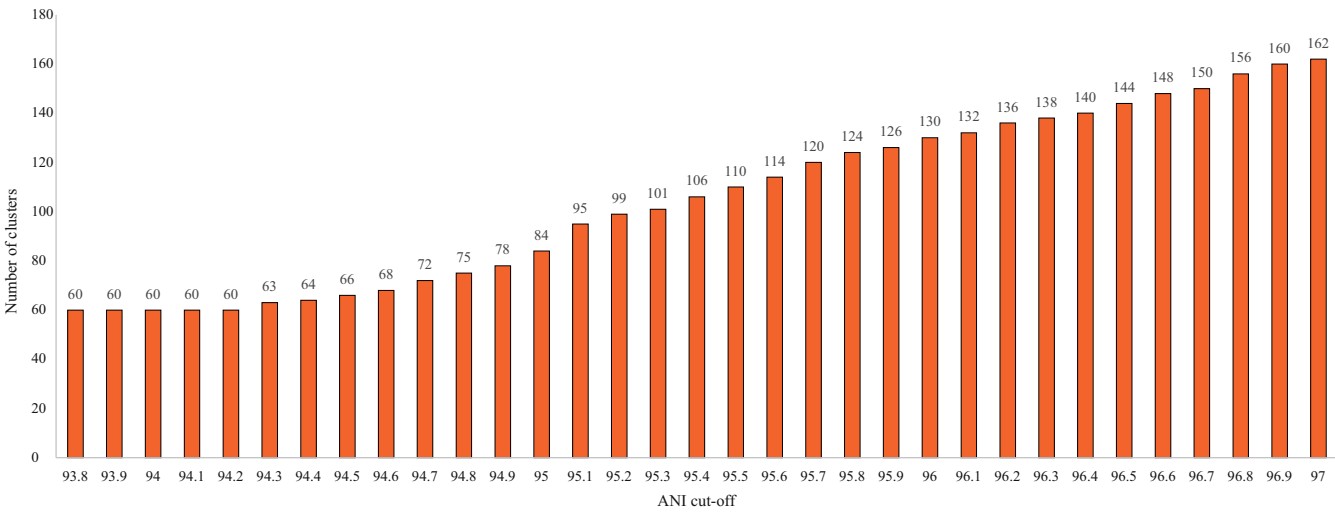

**FIG 2** Hierarchical average linkage clustering of 8,440 genomes based on pairwise comparison of ANI dissimilarity. The X axis shows the ANI cutoff from 93.8% to 97%, while the Y axis shows the corresponding number of ANI clusters.

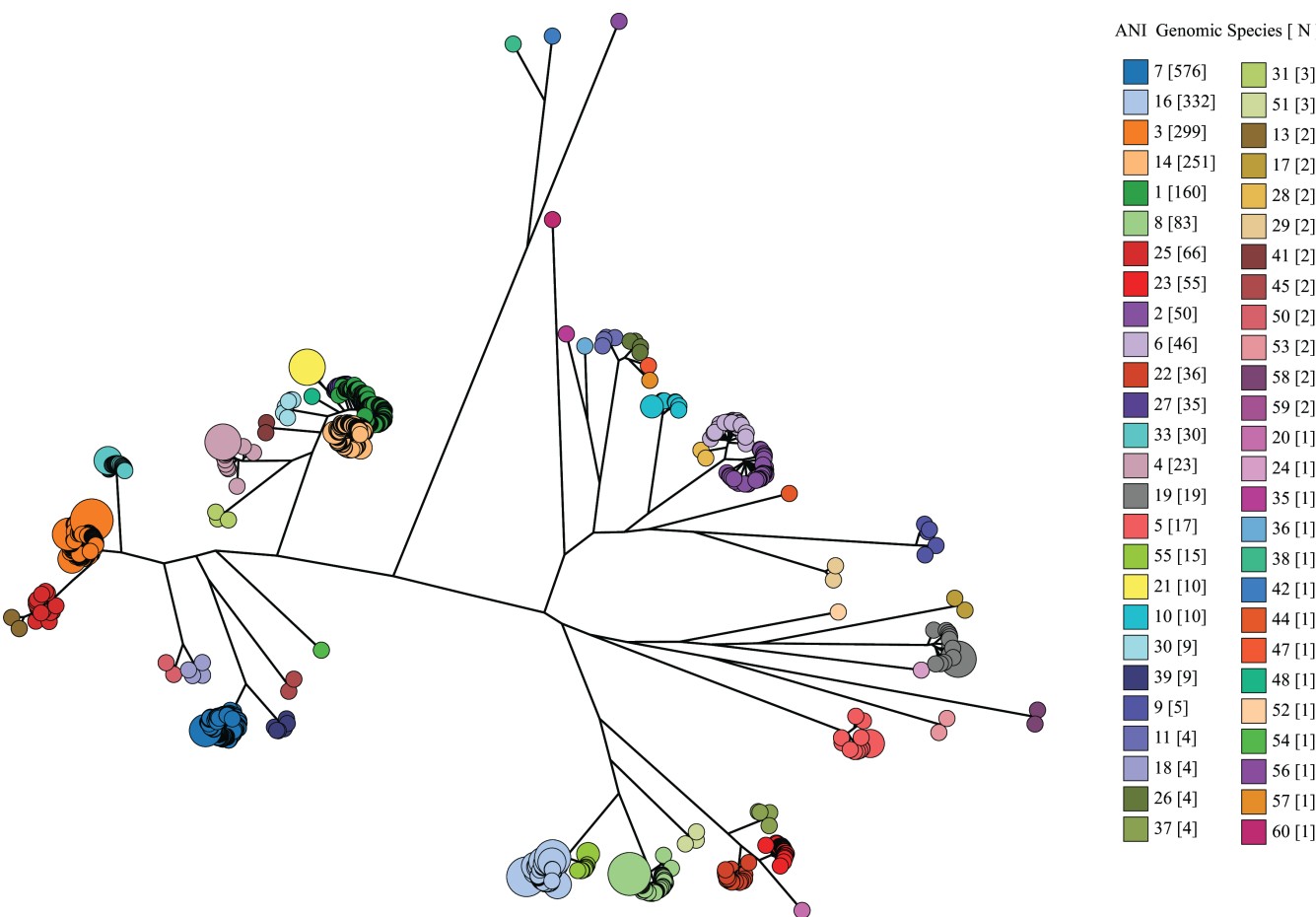

**FIG 3** Core genome phylogenetic tree of 2,193 genomes overlaid with the 60 ANI genomic species. The phylogenetic tree in Fig. 1 was overlaid with ANI genomic clusters. The ANI genomic species were obtained from average linkage clustering of ANI dissimilarity from 8,440 *Campylobacter* genomes at 94.2% ANI. The ANI genomic species (numbered 1–60) and the number of genomes in that species in brackets are as shown in the color legend.

corresponded to *C. lari* subsp. *concheus* and *C. lari* subsp. *lari*, which can be differentiated using cgMLST, ANI, and DDH (40, 46).

## *Campylobacter* ANI genomic species definition

Based on the 60 ANI clusters identified from the 8,440 genomes at ANI 94.2%, we propose these ANI clusters be designated as *Campylobacter* genomic species including 33 established species and 27 novel genomic species (Table S2). A medoid genome in each of the 60 genomic species was identified using PAM algorithm (37) and was used as a reference for each genomic species, respectively. A core genome phylogenetic tree was constructed using the 60 reference genomes and annotated with both NCBI user submitted species and ANI genomic species information in Fig. 4.

## Concordance of ANI clusters with digital DNA-DNA hybridization as genomic species

DDH has been used as the gold standard for genomic comparisons between strains from different species with a similarity threshold of less than 70% as species demarcation. DDH is used as a parameter to define a new species. DDH values were calculated *in silico* between medoid genomes of each ANI genomic species as shown in Fig. 5. Every pair of medoid genomes showed a DDH of less than 70%, supporting each ANI cluster being classified as a distinct genomic species.

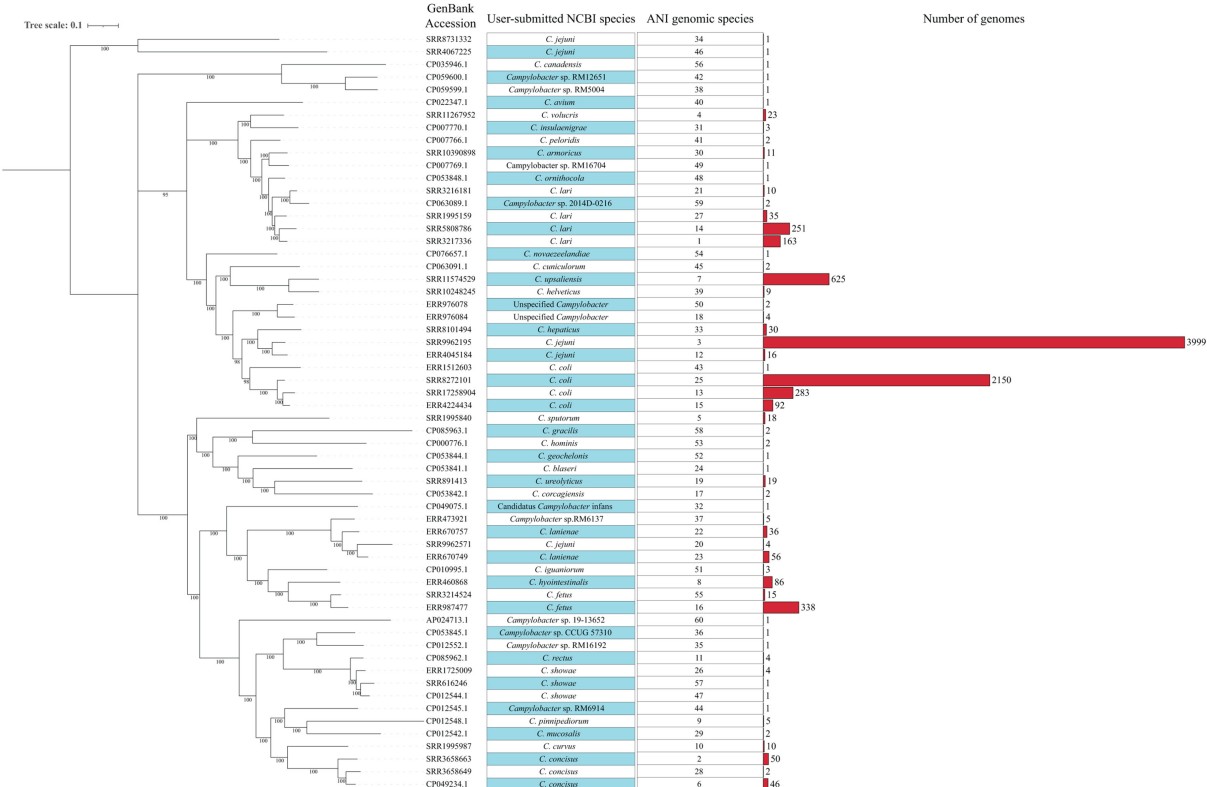

**FIG 4** Core genome phylogenetic tree of medoid genomes of the 60 ANI genomic species. A maximum likelihood phylogenetic tree was generated from the core genome alignments using IQ-TREE v2.0.662 and visualized using iTOL v6.8.1. The bootstrap values indicated at each node were obtained from 10,000 bootstrap replicates and reported as percentages. The phylogenetic tree was visualized in iTOL v6.8.1, and the first column indicates the user submitted species name in NCBI database for the medoid genomes. The second column indicates the corresponding ANI genomic species of each medoid genome. The third column shows the number of genomes belonging to each ANI genomic species in the ANI analysis database (8,440 genomes in total).

## Identification of *Campylobacter* ANI genome species using medoid genomes database

Kraken2 species classification with a default NCBI taxonomy database was unable to consistently assign correct species information to the genomes. From 8,440 genomes in the ANI analysis data set, Kraken2 assigned 5,284 genomes to species that matched with their NCBI user submitted species; 3,007 genomes were assigned to species that were different from their NCBI user submitted species. Additionally, Kraken2 also assigned incorrect species to 149 unspecified *Campylobacter* genomes.

In order to facilitate the assignment of ANI genomic species to new genomes, CampyGStyper was developed. This program utilizes fastANI to compare query *Campylobacter* genomes to the 60 medoid genomes of the ANI genomic species. Assignment of the query genome to a genomic species was based on the highest ANI similarity to a reference medoid genome if it was above 94.2%. In order to detect novel ANI genomic species, *Campylobacter* genomes with less than 94.2% ANI to all of the medoid gnomes will be flagged as potential novel ANI genomic species.

Out of the 8,840 genomes in the ANI analysis data set, 8,438 (99.97%) were assigned correctly to their initially assigned ANI genomic species. Only two genomes were assigned incorrectly. Genome ERR4224450 and SRR5192905 originally belonged to ANI genomic species 25 and 1 and were assigned to ANI genomic species 15 and 27, respectively. ANI genomic species 25 and 15 are all originated from *C. coli* genomes, and ANI genomic species 1 and 27 are all originated from *C. lari* genomes. The ANI of Genome ERR4224450 to ANI genomic species 25 and 15 was 96.26% and 96.35%, respectively, while the ANI of genome SRR5192905 to ANI genomic species 1 and 27 was 93.56%

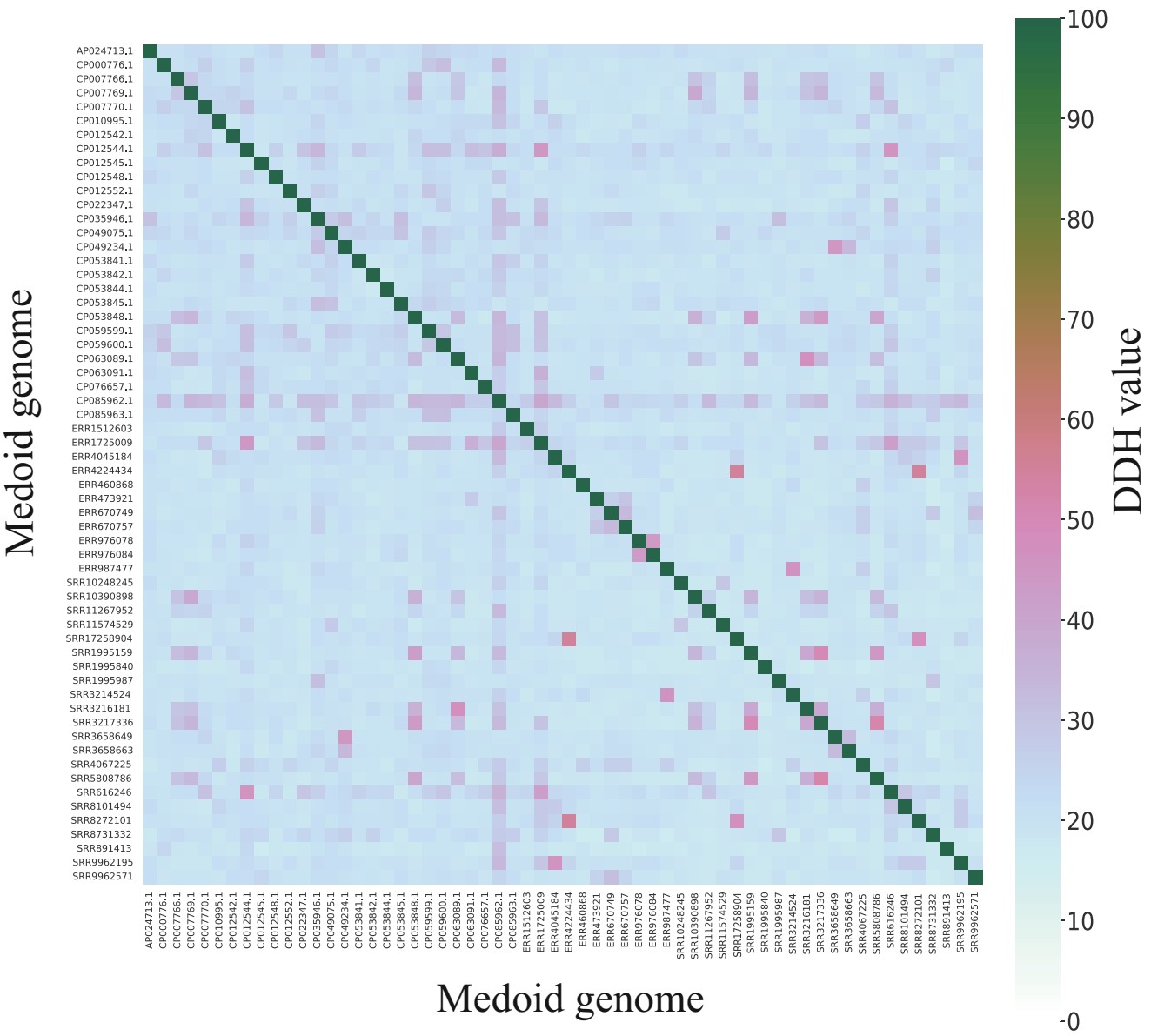

**FIG 5** Pairwise DDH comparison of 60 ANI genomic species medoid genomes. The medoid genome of each ANI genomic species was subjected to pairwise *in silico* DDH comparison using the GGDC web service. The percentage of DDH was shown per the color legend.

and 93.72%, respectively. In both cases, the mis-assignment was due to very similar ANI differences to both genomic species medoid genomes. For Genome ERR4224450, the ANI to both ANI genomic species was below the cutoff of 94.2% ANI, and Genome ERR4224450 was flagged as a potential novel species.

## DISCUSSION

The genus *Campylobacter* currently comprises 33 species with some species further divided into subspecies or genomospecies (1, 3–5). Species definitions were based on a variety of methods including traditional taxonomy, 16S RNA, or genome sequence. Assignment of known and novel species to genome sequences of *Campylobacter* isolates is a challenge due to the high diversity of *Campylobacter* (3, 47). A phylogeny-based species demarcation or species assignment is ideal because members of one species

share the same common ancestor and meet the phylogenetic species concept (48). In this study, the *Campylobacter* core genome was defined, and a core genome phylogeny was constructed using 2,193 genomes representative of the diversity of the genus. Species boundaries consistent with many established species were identified as well as inconsistencies and potential new species. ANI has emerged as a rapid classifier of bacterial species with a generally agreed 95% cutoff for a separate species. Comparative analysis of the *Campylobacter* core genome phylogeny and ANI dissimilarity clustering of 8,440 representative genomes found a cutoff of 94.2% was better suited to demarcate *Campylobacter* species. We identified 60 ANI clusters that were consistent with the core genome phylogeny and proposed that these constitute the sampled genomic species within the *Campylobacter* genus. Of these genomic species, 33 corresponded to traditional species or subspecies, while 27 genomic species were novel without corresponding traditional species, highlighting the diversity of the *Campylobacter* genus.

## *Campylobacter* genus core genome and core genome phylogeny

In this study, we defined the core genome of the *Campylobacter* genus, consisting of 500 loci. Core genome alignments were extracted from 2,193 publicly available genomes representing 33 NCBI user submitted *Campylobacter* species and eight subspecies. Within the 2,193 genomes, the genus core included 29.6% of the average *Campylobacter* strain genome size (1,706,678 bp).

Establishing the core genome will allow future studies of the pan genome of the genus to help understand the genomic diversity and adaptation of different species. Using the *Campylobacter* genus core genome, we derived a core genome phylogeny for classifying *Campylobacter* isolates. The delineation of species on the core genome phylogenetic tree is consistent in many instances with previous studies, as closely related *Campylobacter* species in established groups can be found within the same major clades of the phylogenetic tree (13, 14, 40, 43, 49–51). With the rapid expansion of the *Campylobacter* genomes and additional diversity from new genomes sequenced, it is inevitable that new *Campylobacter* genomes may not be assignable to an established species or may be inconsistent with traditional species. We identified genomes on the core genome phylogeny that belonged to a species different from their NCBI user submitted species. In specific instances, genomes labeled as *C. fetus*, *C. hyointestinalis*, *C. lari*, and *C. upsaliensis* by NCBI user-submissions are in fact *C. coli* or *C. jejuni* based on core genome phylogenetic analysis. Furthermore, based on the branch length of the genus core genome phylogeny, the diversity within and between species varied substantially. The branch length between the two genomospecies of *C. concisus* is comparable if not longer than the branch length between *C. showae* and *C. rectus*. This indicates that intraspecies diversity may be larger than interspecies diversity and divergence, and the demarcation of species and subspecies is inconsistent across the genus. To develop a robust, standardized species definition for the *Campylobacter* genus, we utilized the clustering of pairwise ANI values to complement the core genome phylogeny for species demarcation.

## Species delineation in *Campylobacter* using ANI

Conventionally, the 95% ANI threshold is used as the universal species boundary to differentiate groups of genomes from different genomic species (21, 22). However, previous studies have shown that many *Campylobacter* species have intraspecies similarity lower than 95% ANI (10, 52–54). Many subspecies or biovars would deserve new species status if 95% ANI was applied. Therefore, the 95% ANI cutoff may not be a suitable cutoff for species delineation in *Campylobacter*. A genus-specific ANI cutoff would be better to delineate *Campylobacter* species consistently. From the analysis of 8,440 genomes, we found that the optimal cutoff was 94.2%. At this cutoff, subspecies that are known to be genetically similar, including *C. fetus* (*fetus* and *venerealis*), *C. pinnipediorum* (*pinnipediorum* and *caledonicus*), and *C. hyointestinalis* (*hyointestinalis* and *lawsonii*), were grouped into distinct, species-level genomic species, consistent with

previous studies (53, 55, 56). The biovars of *C. sputorum* belong to a single distinctive ANI cluster compared to the rest of the genus, consistent with their core genome phylogeny and a previous study (54). Our determination of the optimal cutoff was based on both the minimization of ANI clusters in concordance with core genome phylogeny and consistency with predefined species and phylogenetic clusters; thus, the genomic species defined reflect the core genome phylogeny and met the phylogenetic definition of a species. The validity of the 60 ANI genomic species delineation was further confirmed using *in silico* DDH. Traditional species definition uses a DDH of <70% to define a species (36). A 70% DDH was previously shown to correlate well with species boundary at 93%–96% ANI (57, 58).

## Delineation of *C. jejuni*, *C. coli*, and *C. hepaticus*

Unsurprisingly, *C. jejuni* and *C. coli* genomes occupy a common clade on the core genome phylogenetic tree. The similarity between the two species is well-studied, and there are even reports of hybrid isolates caused by *C. jejuni* gene introgression into the *C. coli* gene pool (14, 40). The introgression of the two species is evident on the core genome phylogenetic tree, where a few *C. coli* and unspecified *Campylobacter* genomes are sporadically distributed between the main *C. jejuni* and *C. coli* branches. Core genome phylogenetic analysis alone could not justify including the hybrid genomes to either species or define them as novel genomic species. Using ANI genomic species definition, the majority of the *C. jejuni* and *C. coli* genomes were grouped into ANI genomospecies 3 and 25. The size of these two genomic species aligns with the large populations of the two classic species in the NCBI database. Seven novel ANI genomic species were identified from genomes originally identified as *C. jejuni* (ANI genomic species 12, 20, 34, and 46) and *C. coli* (ANI genomic species 13, 15, and 43). Most notably, the three *C. coli*-originated ANI genomic species are consistent with the established clades of *C. coli* in previous studies, including ANI genomic species 13 and 15, which are two novel genomic species diverged from *C. coli* (59, 60). *C. hepaticus* (ANI genomic species 33) is closely related to the ANI genomic species of *C. jejuni* and *C. coli* (3 and 25, respectively) based on the core genome phylogenetic analysis. The close core genome phylogenetic relationship among the ANI genomic species originated from these three species is consistent with previous studies (49, 61–63). ANI genomic species derived from the three species formed a distinct clade compared to the rest of the genus.

## Delineation *C. lari* group

The previously defined *C. lari* group of species consists of *C. lari*, *C. armoricus*, *C. insulaenigrae*, *C. volucris*, *C. subantarcticus*, *C. ornithocola*, and *C. peloridis* (4, 40). These diverse species can cause human gastroenteritis and are commonly isolated from coastal animals, such as wild birds, shellfish, and marine mammals, as well as seawater (4, 40). ANI genomic species derived from *C. lari* group are phylogenetically close to each other and collectively occupy a distinct clade in the *Campylobacter* genus, consistent with the previous studies (4, 40). Each species-level nomenclature in this group correlates to a distinct ANI genomic species. The two subspecies of *C. lari* correspond to two phylogenetically similar but distinct ANI genomic species (1 and 14), which is consistent with previous studies utilizing cgMLST (4). The smallest of the three *C. lari*-derived ANI genomic species is ANI genomics species 27 with 35 genomes, which represents a novel species in the *C. lari* group. Most of the *C. subantarcticus* genomes were grouped into ANI genomic species 21. The only exception was genome SRR15325077, which was grouped with CP063089.1 in ANI genomic species 59, another novel species in the *C. lari* group.

## Delineation *C. concisus*

*C. concisus* has two well-recognized genomospecies that are genetically distinct from each other (43, 45, 60). The core genome phylogeny and ANI clustering at 94.2% both confirmed that *C. concisus* can be differentiated into two genomic species. *C.*

*concisus* genomospecies 1 and 2 exclusively correspond to ANI genomic species 2 and 6, respectively. DDH between representatives of ANI genomic species 2 and 6 was also less than 70%. The genomospecies classification based on core genome analysis proposed by Wang *et al.* correlates exactly with the ANI genomic species assignment (43). Two *C. concisus* genomes from unknown infection sources formed a novel third *C. concisus* ANI genomic species (28) that is closely related to the ANI genomic species 2 and 6.

## Delineation of *C. fetus*-related non-thermotolerant *Campylobacter* species

The non-thermotolerant *C. fetus*-related *Campylobacter* species includes *C. fetus*, *C. lanienae*, *C. hyointestinalis*, and *C. iguaniorum*. These species are frequently isolated from domestic livestock and can cause illness in both humans and animals (50). The species in this group are all grouped together in a common clade, separate from the rest of the genus on the core genome phylogenetic tree. Each species corresponded to a distinct ANI genomic species. However, the classification of subspecies based on ANI clustering varied between species.

Among the three subspecies of *C. fetus*, subspecies *fetus* and *venerealis* are genomically very similar, and both are known pathogens in cattle (8, 41), whereas subspecies *testudinum* is mostly hosted by reptiles (55). The subspecies status for *fetus* and *venerealis* was based on slight difference in epidemiology and pathological disparities rather than genomic dissimilarity (64). The two subspecies were grouped together into ANI genomic species 16, while subspecies *testudinum* was separated as ANI genomic species 55.

The two subspecies of *C. hyointestinalis*, subsp. *hyointestinalis*, and *Lawsonii*, were grouped into ANI genomic species 8. The two *C. hyointestinalis* subspecies are known to undergo frequent recombination, and the genetic differences between the two subspecies are likely due to host-associated evolutionary events (56).

*C. iguaniorum* corresponds to ANI genomic species 51, which is closely related to ANI genomic species 55 (*C. fetus* subspecies *testudinum*). *C. iguaniorum* and *C. fetus* subspecies *testudinum* are the only two *Campylobacter* taxa isolated from reptiles. It was previously suggested that the similarity was likely due to accumulated host-specific adaptations (55).

The four *C. lanienae*-related putative taxa described by Miller et al. (50) correlated with ANI genomic species 20, 22, 23, and 37. These four ANI genomic species formed a *C. lanienae* clade within the *C. fetus* group based on the core genome phylogeny of the medoid genomes, consistent with the previous study (50).

## The future of *Campylobacter* species delineation and species naming

In the age of easily accessible whole-genome sequencing, subspecies-species definitions lack practical application in clinical settings (65, 66). Subspecies definitions vary from species to species, and different studies define a subspecies using different biochemical test standards (8). The application of a unified genomic-based method to delineate *Campylobacter* species will eliminate the current inconsistencies in *Campylobacter* species and subspecies definitions. ANI genomic species offers a unified nomenclature and standardized species definition for the *Campylobacter* genus.

The traditional species delineation requires phenotypic markers such as biochemical and morphological differences and is, therefore, a challenge when novel species lack these differences. Given that genome sequence may be the first data as the evidence of a new species and genome sequencing is readily available and cost effective, the requirement of phenotypic traits to define a species can hinder both the delineation of new species and removal of inconsistencies in already defined species.

We recommend the adoption of ANI genomic species established in this study to delineate *Campylobacter* species. ANI clustering is well-established and recognized for its application in microbial species definition. An advantage of using ANI genomic species is consistent classifications of subspecies while simultaneously describing established intraspecies genetic divergence. For instance, genetically similar subspecies of *C. fetus*, *C. hyointestinalis*, and *C. pinnipediorum* are consolidated into species-level nomenclature,

consistent with existing species definitions (8, 52, 53). Application of ANI genomic species also removes the inconsistencies and arbitrariness of subspecies or genomo-species, particularly those that are diverse enough to be assigned their own genomic species, such as *C. fetus* subspecies *testudinum* and the two genomospecies of *C. concisus*. The delineation of the *Campylobacter* genus based on ANI genomic species is largely consistent with previously defined *Campylobacter* "community species" using network analysis (60). However, there is no standardized method for network analysis to define new species and type newly defined species. In contrast, the ANI genomic species approach can define new species easily using standardized ANI clustering and ANI cutoff, offering consistent definition of novel species. Newly sequenced genomes can be quickly typed and compared using standardized ANI metrics.

## ANI genomic species assignment and applications

Genomic species assignment using medoid genomes has been applied to the *Bacillus cereus* group by Carroll et al. with high accuracy (67). We have identified the medoid genome for each of the 60 ANI genomic species, which allows one to compare the ANI similarity of unspecified *Campylobacter* genomes against the medoid genomes to determine their ANI genomic species identity. To facilitate this analysis, we present CampyGStyper; a program that assigns *Campylobacter* ANI genomic species to query genomes based on fastANI comparisons with the medoid genomes. The genomic species assignment error rate of CampyGStyper in our data set of 8,840 genomes was only 0.03% (two genomes). The mis-assignment of genomes ERR4224450 and SRR5192905 could be attributed to the frequent horizontal gene transfer between closely related *Campylobacter* species. We reviewed the fastANI output and found that medoid genomes with the second highest ANI similarity to Genome ERR4224450 and SRR5192905 are the correct medoid genomes for ANI genomic species 25 and 1, respectively. CampyGStyper provides fastANI results for users to check when unexpected results are found.

When compared with Kraken2 with a default NCBI taxonomy database, CampyG-Styper provided far superior species classification accuracy. It also utilizes the *Campylobacter*-specific species cutoff at ANI 94.2% to facilitate the identification of novel species. CampyGStyper performs both ANI genomic species assignment and identification of potential new ANI genomic species using the genus-specific ANI cutoff.

There are other tools such as rMLST for *Campylobacter* species identification (68, 69). However, the species identified using rMLST is the same as the existing species designations and does not report potential new species. Hierarchical Bayesian clustering (fastBAPS) can also be used to divide genomes into clusters at different levels, some of which correspond to species divisions (70). We used fastBAPS to analyze the 2,193 genomes of the core genome data set and identified some clusters at different levels that were consistent with ANI genomic species (data not shown). However, fastBAPS also clustered divergent genomes with different ANI genomic species into the same cluster, inconsistent with the phylogenetic clustering and ANI genomic species.

A well-defined ANI genomic species classification for *Campylobacter* is important for identification, diagnosis, clinical management, and epidemiological studies (65). It can facilitate a better understanding of the mechanisms of pathogenesis of human and animal pathogens and the development of targeted therapies or vaccines for prevention.

## Conclusions

In this study, we defined the *Campylobacter* genus core genome and used core genome phylogenetic analysis combined with ANI clustering to delineate *Campylobacter* species. We identified an optimal ANI cutoff of 94.2% rather than the generic 95% for a stand-ardized genomic species delineation and identified 60 ANI genomic species with 33 corresponding to established species and 27 novel species. We also identified the medoid genome for each ANI genomic species to enable rapid identification of a *Campylobacter* ANI genomic species. We expect the proposed ANI genomic species will

facilitate the standardization of species definition in *Campylobacter* genus, enhancing our understanding of the *Campylobacter* species diversity, improving surveillance of established pathogens and facilitating evolutionary and epidemiological studies of novel *Campylobacter* species.

## ACKNOWLEDGMENTS

The authors would like to thank Duncan Smith and Robin Heron from the UNSW ResTech Technology Services Team for their ongoing assistance with high-powered computing and data management systems.

R.W.: collected, processed, and analyzed the data. Constructed the python script used in this project, writing, and editing of the manuscript. M.P.: supervised the project and editing of the manuscript. L.Z.: conceived the project with R.L. and provided feedback on the manuscript. R.L.: conceived and supervised the project and writing and editing of the manuscript.

## AUTHOR AFFILIATION

[1]School of Biotechnology and Biomolecular Sciences, University of New South Wales, Sydney, New South Wales, Australia

## AUTHOR ORCIDs

Ruochen Wu ⓘ https://orcid.org/0000-0002-3500-4517
Michael Payne ⓘ http://orcid.org/0000-0003-1911-7033
Li Zhang ⓘ http://orcid.org/0000-0001-7506-8278
Ruiting Lan ⓘ http://orcid.org/0000-0001-9834-5258

## ADDITIONAL FILES

The following material is available online.

### Supplemental Material

**Figure S1 (mSystems01218-23-s0001.pdf).** Core genome phylogenetic tree comparison with 95% ANI clustering.
**Supplemental tables (mSystems01218-23-s0002.docx).** Tables S1 and S2.

### Open Peer Review

**PEER REVIEW HISTORY (review-history.pdf).** An accounting of the reviewer comments and feedback.

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
