## [Reviewer comments · mSystems]

Uncovering the Boundaries of *Campylobacter* Species through large scale phylogenetic and nucleotide identity Analyses

Ruochen Wu, Michael Payne, Li Zhang, and Ruiting Lan

Corresponding Author(s): Ruiting Lan, University of New South Wales

Review Timeline:

Submission Date:	November 14, 2023
Editorial Decision:	January 11, 2024
Revision Received:	February 13, 2024
Accepted:	February 21, 2024

Editor: Juliette Hayer

Reviewer(s): The reviewers have opted to remain anonymous.

Transaction Report:

DOI: <https://doi.org/10.1128/msystems.01218-23>

Re: mSystems01218-23 (Uncovering the Boundaries of *Campylobacter* Species through large scale phylogenetic and nucleotide identity Analyses)

Dear Prof. Ruiting Lan:

According to the reviews from the reviewers, I have decided to accept this manuscript for publication, with minor modifications. Please take into account the reviewers valuable comments, and make the modifications accordingly.

Revision Guidelines

Sincerely,
Juliette Hayer
Editor
mSystems

Reviewer #1 (Comments for the Author):

In this manuscript the authors use publicly available data to interrogate species boundaries in the genus *Campylobacter*. Several *Campylobacter* species are important human and veterinary pathogens.

As part of the study the authors also provide a useful bioinformatic tool that can be used for clustering genomes and set appropriate nucleotide similarity cut-offs for species designation.

The manuscript is reasonable well written and presented, however I felt there needed to be a clearer aim.

Specific comments

The study is similar to several recent studies on nucleotide similarities in the *Campylobacter* genus, and it would be interesting to see how the results of this work compares (e.g. PMID: 35191377, PMID: 37105244 & PMID: 36069574).

Are there any practical implications to inappropriate *Campylobacter* species designation? E.g. would incorrect species identification hinder therapeutic treatment in any way?

It would be nice to see some comparisons with the range of different species/genome clustering methods already used in other studies. rMLST is used by pubMLST for species designation. Do hierarchical BAPS groups also reconstruct the same genom-species?

Minor

L62 genome content

L138-9: Please describe briefly the previous Python script

L182-190: How does this imputed multi species core genome compare with those inferred in other studies?

Reviewer #2 (Comments for the Author):

In this study, the authors use publicly available *Campylobacter* genomes to define core genes, construct a core genome phylogeny, and to delineate species by refine an ANI cut-off. Overall, the work provides greater clarity for the classification of species within the *Campylobacter* genus.

General comments:

It is sometimes difficult to follow the exact number of genomes used for different purposes throughout the manuscript. Perhaps an explanatory table could be helpful.

Supplementary Figure 1 seem to not be needed and does not add much to the work.

It is interesting the the *C. lari* group is referred to as a 'group'. In the context of ANI, what does 'group' mean and was there any change to this definition based on ANI? Are the other species that should be thought of as groups or complexes based on your ANI results?

Specific comments:

Line 19. Suggest change to 'typically helical shaped'.

Line 58. Suggest 'domesticated'.

Line 61-62. Recommend rewording gene content statement more in line with an evolutionary process.

Line 81. Delete 'High Level of'.

Line 85-85. Suggest clarifying this sentence about MLSTs found among different *Campylobacter* sequences.

Line 306. Suggest 'comprises'

Line 309. Suggest changing 'have become' to 'is'.

Line 433. This sentence is difficult to understand.

Table 1. Although I understand why there may decimal places, I think it would make more sense to either present all numbers with the same decimals/significant figures? Or round up to the nearest full number.

Supplementary Figure 1. Typo "ANI Clucter".

We thank the editor for handling our manuscript and the reviewers for providing constructive comments that improved the manuscript. Please find our responses to comments point by point below. Some other minor changes include rename our typing tool to CamplyGStyper. The changes we have made in the manuscript are indicated in bold.

Reviewer #1 (Comments for the Author):

*In this manuscript the authors use publicly available data to interrogate species boundaries in the genus *Campylobacter*. Several *Campylobacter* species are important human and veterinary pathogens.*

As part of the study the authors also provide a useful bioinformatic tool that can be used for clustering genomes and set appropriate nucleotide similarity cut-offs for species designation. The manuscript is reasonable well written and presented, however I felt there needed to be a clearer aim.

Specific comments

*The study is similar to several recent studies on nucleotide similarities in the *Campylobacter* genus, and it would be interesting to see how the results of this work compares (e.g. PMID: 35191377, PMID: 37105244 & PMID: 36069574).*

Our responses: We have compared our result with the publications provided by reviewer 1. The three established clades of *C. coli* correlated with the three *C. coli* ANI genomic species in our study. Our result has also shown *C. concisus* genomospecies 2 is distinctively different from genomospecies 1 as two ANI genomic species.

PMID 35191377 and 36069574 used core genome phylogeny to examine relationships of the *Campylobacter* species which is similar to our core genome phylogenetic analysis albeit we have a larger set of genomes with a better-defined genus core. The phylogenies are consistent across studies for

1. The close relationship between *C. jejuni*, *C. coli* and *C. hepaticus*.
2. The distinct phylogenetic positions of the species in the previously defined *C. lari* group
3. The distinct phylogenetic positions of *C. fetus* and related species.
4. The close relationship of *C. showae* and *C. rectus*

The following changes have been made to the manuscript with new references added.

Line 390-391: Most notably, the three *C. coli* originated ANI genomic species are consistent with the established clades of *C. coli* in previous studies, including ANI genomic species 13 and 15, which are two novel genomic species diverged from *C. coli* (58, 59).

Line 415-416: *C. concisus* has two well-recognized genomospecies that are genetically distinct from each other (43, 45, 59)

PMID 37105244 reported novel species through network analysis. The approach separates genomes into “communities” which are equated to species. They had proposed Clade III of *C. coli* as a novel species (referred here as *Campylobacter* spp12) and proposed an additional 43 novel species. We compared these novel species with ours. They are largely consistent. We used the genome accession numbers from PMID 37105244 to type them using our CampyGStyper. Note that PMID 37105244 used assembled genomes from GenBank. Seven community species in PMID 37105244 were not defined in our ANI genomic species and thus potentially new species. Four community species genomes were not typable using our ANI genomic species typer due to genome assembly anomalies as marked by GenBank and no longer retrievable. Ten ANI genomic species we defined were not found among the community species defined by PMID 37105244 as it does not have a typing tool to type these genomes which were assembled by us from raw reads.

However, the approach used is only used in the literature for a couple of studies compared to the ANI approach we used which is well established and recognised. Further we provide a tool that can type and potentially define new species consistently. The network analysis approach has no standardised method or tool to further type or define new species.

We have added the following to the manuscript:

Line 474: ANI clustering is well-established and recognized for its application in microbial species definition.

Line 481: The delineation of the *Campylobacter* genus based on ANI genomic species is largely consistent with previously defined *Campylobacter* “community species” using network analysis (60). However, there is no standardised method for network analysis to define new species and type newly defined species. In contrast, the ANI genomic species approach can define new species easily using standardised ANI clustering and ANI cut-off, offering consistent definition of novel species. Newly sequenced genomes can be quickly typed and compared using standardised ANI metrics.

Are there any practical implications to inappropriate Campylobacter species designation? E.g. would incorrect species identification hinder therapeutic treatment in any way?

Our response:

Appropriate designation of a species is important for diagnosis, identification, and epidemiological studies. It can facilitate better understanding of the mechanisms of pathogenesis of human and animal pathogens, development of targeted therapies or vaccines for prevention.

For example, infections in human caused by emerging zoonotic pathogens such as *C. fetus* and *C. lanienae* are more prevalent in recent years due to the increased international movement of supplies and livestock. The pathogenicity of these less common pathogens in humans is not as well understood compared to established human pathogens such as *C. jejuni* and *C. coli*. Accurate species identification is crucial to prevent misidentification in clinical samples, therefore improving the quality of epidemiology studies related to these pathogens.

We have added the following sentences in the manuscript:

Line 503: A well-defined ANI genomic species classification for *Campylobacter* is important for identification, diagnosis and clinical management of human infections, and epidemiological studies (72). It can facilitate better understanding of the mechanisms of pathogenesis of human and animal pathogens, development of targeted therapies or vaccines for prevention.

It would be nice to see some comparisons with the range of different species/genome clustering methods already used in other studies. rMLST is used by pubMLST for species designation. Do hierarchical BAPS groups also reconstruct the same genomo-species?

Our response:

The rMLST species identification algorithm is based on allele matches with existing genomes and report species as defined by NCBI species designations. Therefore, rMLST is not a species defining or new species identification tool. It's a tool that can type to existing species defined. Nevertheless, we have used rMLST in PubMLST database to type the species for the medoid genomes of each ANI genomic species. The rMLST database does not differentiate established variants of traditional *Campylobacter* species. For example, medoid genomes of the three *C. coli* originated ANI genomic species are all identified as *C. coli*. The species identified by the rMLST approach for all medoid genomes of each ANI genomic species were the same

as the NCBI species designation. Therefore, it is inappropriate to compare our approach with rMLST species identification tool.

We used the 2193 core genome alignments to run fastBAPS (PMID: 31076776) with four levels. At levels 1-4, 17, 31, 40 and 46 clusters were identified respectively. There were clusters at each level that were fully or partly consistent with our ANI genomic species. However, the worst BAPS clustering outcome was the clustering of divergent and rare *Campylobacter* genomes/species into the same cluster as shown in figure below. It seems this behaviour is analogous to long branch attraction in phylogenetic analysis. BAPS is a good population structure analysis tool. However, we believe it is a poor tool for classifying species and defining new species. Further even if BAPS can cluster species sensibly, it cannot be easily used as a species typing tool. Given the poor performance of BAPS and no previous BAPS analysis of *Campylobacters*, we decided not to incorporate our BAPS analysis results into the manuscript but only briefly mentioned the approach in the MS.

Figure caption: The core genome alignments are labelled with their ANI cluster numbers. The fastBAPS level 4 clusters were overlaid on the phylogenetic tree and indicated with colour legend. fastBAPS level 4 cluster 37 (in red) included core genome alignments from multiple ANI clusters. The core genome alignments in cluster 37 are also scattered across the phylogenetic tree.

Line 497: There are other tools such as rMLST for *Campylobacter* species identification (69, 70). However, the species identified using rMLST is the same as the existing species designations and doesn't report potential new species. Hierarchical Bayesian clustering (fastBaps) can also be used to divide genomes into clusters at different levels, some of which correspond to species divisions (71) . We used fastBAPS to analyse the 2193 genomes of the core genome dataset and identified some clusters at different levels that were consistent with ANI genomic species (data not shown). However, fastBAPS also clustered divergent genomes/ANI genomic species into the same cluster, inconsistent with the phylogenetic clustering and ANI genomic species.

Minor

L62 genome content

Our response: Sentence has been revised for readability.

Line 61-62. Genomic analyses of *Campylobacter* from different animal reservoirs have demonstrated that *Campylobacter* species achieve host-specific adaptations through horizontal gene transfer (1, 9, 10).

L138-9: Please describe briefly the previous Python script

Our response: The sentence has been revised and included description identification criteria of the python script.

L138-9: Core genes that are paralogous and orthologous split into multiple fragments by erroneous assemble and/or annotation were identified and subsequently removed using a previously described Python script (30).

L182-190: How does this imputed multi species core genome compare with those inferred in other studies?

Our response: The core genome phylogenetic tree constructed in this study is consistent with previous phylogenetic studies. As mentioned above, we have compared with the *Campylobacter* core genome phylogenetic study (PMID: 36069574) provided and similar intra genus phylogenetic relationships have been found.

PMID 36069574 has been added to our discussion:

Line (336-339): The delineation of species on the core genome phylogenetic tree is consistent in many instances with previous studies, as closely related *Campylobacter* species in established groups can be found within the same major clades of the phylogenetic tree (13, 14, 40, 43, 49–51).

Reviewer #2 (Comments for the Author):

In this study, the authors use publicly available Campylobacter genomes to define core genes, construct a core genome phylogeny, and to delineate species by refine an ANI cut-off. Overall, the work provides greater clarity for the classification of species within the Campylobacter genus.

General comments:

It is sometimes difficult to follow the exact number of genomes used for different purposes throughout the manuscript. Perhaps an explanatory table could be helpful.

Our response: A supplementary table has been added to indicate the composition of datasets used in this study.

Supplementary Figure 1 seem to not be needed and does not add much to the work.

Our response: Supplementary figure 1 was included to showcase the 95% ANI cut-off divide campylobacter species into numerous small clusters. Circles as been added to indicate the species being influenced the most and figure legend was updated. We believe this figure is useful to readers who are interested to see the clusters using the traditional 95% ANI cut-off.

It is interesting the the C lari group is referred to as a 'group'. In the context of ANI, what does 'group' mean and was there any change to this definition based on ANI? Are the other species that should be thought of as groups or complexes based on your ANI results?

Our response: The *C. lari* group was coined by previous studies to describe species closely related to *C. lari* but phylogenetically distinct from the rest of the genus. These species are grouped together based on similarity in metabolic pathways or isolation sources. Similarly, previous studies have also described a group of *C. fetus* related non-thermotolerant species being a distinctive clade within the genus *Campylobacter*. The core genome phylogenetic tree constructed in this study is consistent with the previous studies, as both the *C. lari* group and the *C. fetus* related non-thermotolerant species occupies a distinctive clade on the core genome polygenetic tree.

We defined an optimal *Campylobacter* species cut-off at 94.2%. ANI clusters produced at this cut-off is consistent with previous established species and subspecies borders. However, because the “group” concept is not consistent within the genus, it is difficult to define “group” by a cut-off and we have no intention to introduce another layer of division.

We have changed the wording to explicitly state: The previously defined *C. lari* group of species and added references to the sentence so it doesn't cause any misunderstanding.

Line 399-400: The previously defined *C. lari* group of species consists of *C. lari*, *C. armoricus*, *C. insulaenigrae*, *C. volucris*, *C. subantarcticus*, *C. ornithocola* and *C. peloridis* (4, 40).

Specific comments:

Line 19. Suggest change to 'typically helical shaped'.

Our response: Sentence has been corrected.

Line 19: *Campylobacter* species are typically helical shaped, Gram-negative, and non-spore forming bacteria.

Line 58. Suggest 'domesticated'.

Our response: Sentence has been corrected.

Line 58: *Campylobacter* species are diverse and naturally colonise humans, domesticated animals (such as dogs, cats, chickens, sheep, and cattle)

Line 61-62. Recommend rewording gene content statement more in line with an evolutionary process.

Our response: Sentence has been rewritten for readability.

Line 61-62. Genomic analyses of *Campylobacter* from different animal reservoirs have demonstrated that *Campylobacter* species achieve host-specific adaptations through horizontal gene transfer (1, 9, 10).

Line 81. Delete 'High Level of'.

Our response: Sentence has been corrected.

Line 81: Horizontal gene transfer (HGT) between *Campylobacter* species also poses challenges to defining the species boundaries within this genus (10, 13, 14).

*Line 85-85. Suggest clarifying this sentence about MLSTs found among different *Campylobacter* sequences.*

Our response: Sentence has been rewritten for readability.

Line 85-86: Techniques such as multi-locus sequence typing (MLST) are compromised because the extent of recombination between species is so broad, for example, *C. jejuni* sequences can be found in *C. coli* housekeeping genes, making it difficult to differentiate the two species using MLST(14–16).

Line 306. Suggest 'comprises'

Our response: Sentence has been corrected.

Line 306: The genus *Campylobacter* currently comprises 33 species with some species further divided into subspecies or genomospecies (1, 3–5).

Line 309. Suggest changing 'have become' to 'is'.

Our response: Sentence has been corrected.

Line 309: Assignment of known and novel species to genome sequences of *Campylobacter* isolates is a challenge due to the high diversity of *Campylobacter* (3, 47).

Line 433. This sentence is difficult to understand.

Our response: Sentence has been rewritten for readability.

Line 433-434: Among the three subspecies of *C. fetus*, subspecies *fetus* and *venerealis* are genomically very similar and both are known pathogen in cattle (8, 41), whereas subspecies *testudinum* is mostly hosted by reptiles (54).

Table 1. Although I understand why there may decimal places, I think it would make more sense to either present all numbers with the same decimals/significant figures? Or round up to the nearest full number.

Our response: All numbers are now presented with one decimal place.

Supplementary Figure 1. Typo "ANI Clucter".

Our response: The typo has been corrected to “ANI Clusters”

Re: mSystems01218-23R1 (Uncovering the Boundaries of *Campylobacter* Species through large scale phylogenetic and nucleotide identity Analyses)

Dear Prof. Ruiting Lan:

Thank you for addressing all the reviewers questions and comment, I am happy to accept your manuscript for publication in mSystems.

As your manuscript has been accepted, I am forwarding it to the ASM production staff for publication. Your paper will first be checked to make sure all elements meet the technical requirements. ASM staff will contact you if anything needs to be revised before copyediting and production can begin. Otherwise, you will be notified when your proofs are ready to be viewed.

Cover Image Submissions: If you would like to submit a potential Cover Image, please email a file and a short legend to msystems@asmusa.org. Please note that we can only consider images that (i) the authors created or own and (ii) have not been previously published. By submitting, you agree that the image can be used under the same terms as the published article. Image File requirements: TIF/EPS, 7.5 inches wide by 8.25 inches tall (at least 2,250 pixels wide by 2,475 pixels tall), minimum 300 dpi resolution (600 dpi preferred), RGB, and no figure elements, e.g., arrows or panel labels. The legend should be a short description of the image, 1-2 sentences recommended.

We recognize that the video files can become quite large, so to avoid quality loss ASM suggests sending the video file via <https://www.wetransfer.com/>. When you have a final version of the video and the still ready to share, please send it to mSystems staff at msystems@asmusa.org.

Sincerely,

Juliette Hayer
Editor
mSystems

Reviewer #1 (Comments for the Author):

The authors have addressed all concerns raised by the reviewers and incorporated feedback into the a revised version of the manuscript.

Reviewer #2 (Comments for the Author):

The revisions to the manuscript have added clarity. The work is a solid contribution towards the characterization of this important human pathogen.